# Trend Distribution of Violent Injuries in Taiwan from 2000 to 2015

**DOI:** 10.3390/ijerph19137874

**Published:** 2022-06-27

**Authors:** Yao-Ching Huang, Chia-Peng Yu, Bing-Long Wang, Ren-Jei Chung, Iau-Jin Lin, Chi-Hsiang Chung, Chien-An Sun, Pi-Ching Yu, Shi-Hao Huang, Wu-Chien Chien, Sheng-Tang Wu

**Affiliations:** 1Department of Medical Research, Tri-Service General Hospital, Taipei 11490, Taiwan; ph870059@gmail.com (Y.-C.H.); yu6641@gmail.com (C.-P.Y.); iaujinlin@gmail.com (I.-J.L.); g694810042@gmail.com (C.-H.C.); 2School of Public Health, National Defense Medical Center, Taipei 11490, Taiwan; billwang1203@gmail.com; 3Department of Chemical Engineering and Biotechnology, National Taipei University of Technology (Taipei Tech), Taipei 10608, Taiwan; rjchung@ntut.edu.tw; 4Graduate Institute of Life Sciences, National Defense Medical Center, Taipei 11490, Taiwan; 5Taiwanese Injury Prevention and Safety Promotion Association (TIPSPA), Taipei 11490, Taiwan; 6Department of Public Health, College of Medicine, Fu-Jen Catholic University, New Taipei City 242062, Taiwan; 040866@mail.fju.edu.tw; 7Big Data Center, College of Medicine, Fu-Jen Catholic University, New Taipei City 242062, Taiwan; 8Graduate Institute of Medicine, National Defense Medical Center, Taipei 11490, Taiwan; yupichin1003@gmail.com; 9Division of Urology, Department of Surgery, Tri-Service General Hospital, National Defense Medical Center, Taipei 11490, Taiwan

**Keywords:** injury, trend distribution, violence, Taiwan

## Abstract

This study aims to understand the trend distribution of violent injuries in Taiwan from 2000 to 2015. It used the data of outpatient, emergency, and hospitalization of 2 million people in the National Health Insurance sample from 2000 to 2015. We analyzed children and adolescents (hereinafter referred to as children, 0–17 years old), adults (18–64 years old), and The Elderly (over 65 years old) who suffered for the first time. The standardized rate of medical treatment for violent injuries was compared annually using the Poisson regression method. A total of 11,077 victims (7163 men, 3914 women) suffered violence during the 15 years, and the standardized rate of medical treatment for violence in adults dropped from 6.01 (1/10^4^) in 2001 to 2.58 (1/10^4^) in 2015. The standardized rate of medical treatment in adults over the years was higher than that in children (2.96_2001_, 1.23_2015_) and The Elderly (3.52_2001_, 1.62_2015_). The medical treatment rate of the adult generation is higher than that of the children and the elderly. The relative hazard ratio (RR) decreased from 2.38 in 2001 to 1.13 in 2014 (but the RR in 2014 was not significant). Furthermore, the rate of adult violence treatment has been decreasing every year, which shows that the government has achieved remarkable results in general violence prevention. With the accelerated aging of Taiwan’s population, it is expected that older adults exposed to the risk of violence will also increase and become more serious. Therefore, the government should continue to pay attention to this issue.

## 1. Introduction

Violence and injury are important public health issues of global concern. Globally, more than 1 million people die each year due to various types of violence, and many more suffer non-fatal injuries [1]. World Health Organization (WHO) estimates that globally, approximately 470,000 people are victims of violence each year. Hundreds of millions of men, women, and children experience non-lethal interpersonal violence, including child abuse, youth violence, intimate partner violence, sexual violence, and older adult abuse, many of whom suffer in multiple forms [2]. This violence results in lifelong ill-health, particularly in women and children, and in premature death [1,2]. The World Health Organization defines violence as the deliberate threat or actual use of force or power against oneself, another person or group, or community that results in or has a high likelihood of causing injury, death, psychological harm, stunting, or deprivation [3]. Violence can generally be divided into “interpersonal violence” and “collective violence” [3]. Interpersonal violence (IPV) and collective violence are highly likely to result in injury, death, psychological harm, stunting, or deprivation [3]. Worldwide, it is estimated that more than 1.6 million people die from violence (28.8 per 10^5^), and many more are injured and/or suffer from long-term physical or mental health disabilities or conditions [3].

Injuries include unintentional and violent injuries—claiming the lives of 4.4 million people worldwide each year, or nearly 8% of all deaths; for 5–29-year-olds, 3 of the 5 leading causes of death are injury-related, namely road traffic injuries, homicides, and suicides [4]. Injuries and violence burden national economies, costing countries billions of dollars each year in healthcare, lost productivity, and law enforcement [5]. Injuries and violence are significant causes of death and disease in all countries. However, they are unevenly distributed across or within countries—some people are more vulnerable than others, depending on how they were born, raised, worked, lived, and aged [6]. For example, in general, the youth, males, and individuals with a low socioeconomic status increase the risk of injury and being a victim or perpetrator of severe physical violence. Furthermore, the risk of fall-related injuries increases with age [7].

Violence-related harm refers to harm caused by the intentional use of force or power against oneself or others [8]. Interpersonal violence remains the leading cause of violence-related mortality globally (30.5%), with 8.8 per 10^5^ deaths [1]. In 2013, nearly 1.2 million deaths were related to intentional injury, with interpersonal violence (IPV) accounting for 32% and self-directed violence accounting for 68% [9]. Interpersonal violence includes domestic and intimate partner violence (children, partners, or older adults) and community violence (strangers or acquaintances). The estimated lifetime prevalence of intimate partner violence in women in the United States ranges from 28 to 54 percent [10,11]. According to reports from Canada and Switzerland, the prevalence of child sexual abuse ranges from 3 to 40 percent [12,13]. In December 2021, Taiwanese legislator Jia-Yu Gao was accused of being violently injured by her boyfriend, causing Taiwanese society to pay close attention to domestic violence and women’s rights issues. Whether the relevant laws are sufficient to deal with new types of online sex crimes and protect victims has become a topic of public discussion [14]. Workplace violence (WPV) is a serious health care problem in Taiwan, as it is worldwide. Among all nursing staff, emergency department (ED) nurses are at the highest risk of WPV, yet little attention has been paid to nurses as WPV victims [15]. The current preventive measures for WPV against emergency nurses in Taiwan are not effective [16]. Therefore, relevant measures should be improved, thereby reducing the prevalence and severity of workplace violence against emergency nurses [16].

Data are available on the medical burden of violence-related injuries in Western countries. However, there is insufficient research on exposure to violence and injuries in Asia. In Taiwan, there are no published studies on violence-related injuries, especially those requiring hospitalization. Therefore, this study uses the data collected from the Health and Welfare Data Science Center of the Ministry of Health and Welfare (HWDC, MOHW) as a case study to seek medical attention due to violent injuries, to understand the epidemiological characteristics of the trend distribution of violent injuries in Taiwan over the years from 2000 to 2015. We have thereby discussed the crude rate of medical abuse, the standardized rate of medical abuse, and the annual relative hazard ratio for different generations.

## 2. Materials and Methods

### 2.1. Data Sources

Taiwan’s National Health Insurance introduced the single payment system on 1 March 1995. As of 2017, 99.9% of Taiwan’s population participated in the program. This study was a 16-year observational study and used the NHIRD to provide a parent cohort of 2 million people representing the NHIRD 2000 coverage sample (Longitudinal Health Insurance Research Database, LHID 2000) as the study data source. It tracked 20,001 new case data for 16 years from 1 December to 31 December 2015. The documents used are outpatient prescription and treatment detail files, inpatient medical expense list detail files, and insurance information files. It includes 11,077 violent abuse study cases. NIH previously encrypted all personal information.

Release of LHID 2000 to protect patient privacy. In LHID 2000, the diagnostic codes for this disorder are based on the “International Classification of Diseases, Ninth Edition, Clinical Modification” (ICD-9-CM) N-code standard. Cases that occurred in 2000 were excluded. All procedures involving human participants performed in the research were conducted in accordance with institutional and/or national research council ethical standards and the 1964 Declaration of Helsinki and subsequent amendments or similar ethical standards. All methods were performed in accordance with relevant guidelines and regulations. The Ethical Defense Medical Center General Hospital Review Board (C202105014) approved this study.

### 2.2. Participants

Children and adolescents who are defined as victims of violence are minors under the age of 18 enrolled in the National Health Insurance for medical treatment. Scope, as defined in the International Classification of Diseases, Ninth Revision, Clinical Modification (ICD-9 CM) N-code: 995.5 and External Classification Code (E-code): E960–E969, refers to violent abuse of adults including those aged between 18–64. Further, aged violently abused means individuals who are 65 years of age and older, according to ICD-9 N-Code: 995.8 and E-Codes E960–E969, as Case Group (Victim Violence). Injury type (except ICD-9 codes 800–999, 905–909, and 958–958), medical costs ($), and prognosis (survival or mortality).

The CCI selects the top five diagnostic codes (ICD-9-CM N-Code), weights them according to the scoring criteria defined by Charlson and calculates the total score [17]. Higher scores indicate more complications or a more serious diagnosis.

### 2.3. Statistical Analysis

This study was analyzed using the SAS 9.4 statistical software for Windows (SAS Institute, Cary, NC, USA) provided by the Central Research Institute Division of the Data Welfare Center, Department of Health and Welfare. Descriptive statistics were presented as percentages, means, and standard deviations, and chi-square tests were used to compare differences among three groups (children, adults, and older adults). The crude medical treatment rate, the direct standardized medical treatment rate, and the relative risk ratio (RR) of the victims of violence were analyzed. When calculating the crude medical treatment rate, population by gender and age group in each county and city over the years is used. When calculating the standardized medical treatment rate, the world standard population in 2000 was referred to. Poisson regression was used to compare the medical treatment rates of violence each year and each county and city. It was stratified by gender or generation and used Poisson regression to calculate the rate ratio (RR) because over the years, the main body of the medical treatment rate was showing a downward trend. Thus, 2015 is used as the reference year to calculate the ratio of the medical treatment rate in each year relative to 2015, the ratio of medical treatment rates in the city. Differences in the distribution of trends in the history of violent injuries across age groups were identified. According to the central limit theorem, (a) if the sample data is approximately normal, then the sampling distribution will also be normal; (b) in large samples (>30 or 40), the sampling distribution tends to be positive regardless of the shape of the data; (c) the mean of a random sample from any distribution will itself have a normal distribution [7]. A *p*-value < 0.05 was considered statistically significant.

## 3. Results

### 3.1. Annual Medical Treatment Rate

Observing the rough medical treatment rates due to violent injuries in the three generations over the years, the initial medical treatment rates of children and adults who were victims of violence have consistently decreased every year. However, the trend test of the older adult generation showed no significant change. The crude medical treatment rate for adults dropped from 6.04 (1/10^4^) in 2001 to 2.42 (1/10^4^) in 2015. The crude rates of medical treatment for adults were higher than those for children (3.13_2001_, 1.52_2015_) and The Elderly (1.04_2001_, 0.78_2015_). In 2010, there was a sharp increase in the incidence of violent medical treatment in all three generations (Figure 1) (Table 1).

### 3.2. Annual Normalization Rate

The standardized rate of medical treatment for violence in adults dropped from 6.01 (1/10^4^) in 2001 to 2.58 (1/10^4^) in 2015. The standardized rate of medical treatment in adults over the years was higher than that in children (2.96_2001_, 1.23_2015_) and The Elderly (3.52_2001_, 1.62_2015_). In 2010, a sharp increase in the standardized rate of medical treatment for violence was found in the adult and The Elderly cohorts (Figure 2) (Table 2).

### 3.3. Annual Relative Hazard Ratios

After adjusting for the age of first violence, using the standardized rate of medical treatment for violence in 2015 for the reference group, the medical treatment rate of the adult generation was higher than that of the children and older adults. The relative hazard ratio (RR) decreased from 2.38 in 2001 to 1.13 in 2014 (but the RR in 2014 was not significant). Changes in RR among children and adolescents were higher than in 2015, and the RRs from 2001 to 2014 ranged from 1.21 to 1.89. The change in RR in the older generation was lowest in 2005 (RR = 0.86 (95% CI = 0.53–1.40)) and highest in 2010 (RR = 2.54 (95% CI = 1.75–3.67)) (Table 3) (Figure 3). 

Among male victims of violence, a downward trend can be observed in both adults and children. The standardized rate of medical treatment for the elderly began to rise in 2010, overlapping with the standardized rate of medical treatment for adults. In 2007, it began to be higher than the standardized rate of medical treatment for children. After 2010, it highlights that the severity of medical treatment due to violence in the elderly is actually equivalent to that in adults. The standardized rate of medical treatment for adults who suffered from violence dropped from 8.58 (1/10^4^) in 2001 to 2.68 (1/10^4^) in 2015. The standardized rate of medical treatment over the years was higher than that of children (3.41_2001_, 0.63_2015_), and the elderly (3.96_2001_, 2.40_2015_). (Table 4) (Figure 4).

Among female victims of violence, no matter what generation they are, there is no obvious trend change after the test. The lowest to highest standardization rate of medical treatment for children and children was 1.36–3.18 (1/10^4^), the lowest to highest standardization rate for adult medical treatment was 2.36–3.45 (1/10^4^), and the lowest to highest standardization rate for elderly medical treatment. between 0.85–3.35 (1/10^4^) (Table 5) (Figure 5).

This study analyzed the severity of injuries caused by ICD9-based (Appendix A) violence across three generations.

## 4. Discussion

This study describes the distribution of trends in medical treatment in Taiwan from 2000 to 2015 due to factors related to a violent injury. The results determine that the crude rates of medical treatment in adulthood over the years were higher than those in children and older adults, and the standardized rates of medical treatment in adulthood were higher than those in children and older adults. The medical treatment rate of the adult generation is higher than that of children and older adults. The adult relative hazard ratio (RR) decreased from 2.38 in 2001 to 1.13 in 2014 (but the RR in 2014 was not significant). Changes in RR among children and adolescents were higher than in 2015, and the RRs from 2001 to 2014 ranged from 1.21 to 1.89. The change in RR of the older generation was lowest in 2005 (RR = 0.86 (95% CI = 0.53–1.40)) and highest in 2010 (RR = 2.54 (95% CI = 1.75–3.67)).

According to the 2001 Annual Report of the Police Administration of the Republic of China, the number of violent crimes (intentional homicide, robbery, robbery, sexual assault, serious injury, etc.) and the number of victims of violent crimes have dropped significantly in the past 20 years [18]. The results of this study also show that the incidence of violence in children or adults has been consistently decreasing every year, which may be related to the improvement of public security and the reduction of violent crimes. According to a 2020 study in India, the low education level of victims of violence is associated with a higher risk of violence [19]. In addition, a study on domestic violence in Saudi Arabia also noted that both the perpetrator and the victim’s low education level are high-risk factors for domestic violence [20]. In this study, the rough rates of medical treatment for adults over the years were higher than those for young children and older adults. The possible reason is the difference in the population of higher education students. According to the results of the “International Comparison and Analysis of Education Statistical Indicators” published by the Taiwan Ministry of Education in 2015, the proportion of higher education in Taiwan in 2012 was 72% [21]. In an era of low birth rates, promoting high-quality education policies and strengthening the concept of gender equality will help reduce the incidence of violence. In addition, all counties and cities in Taiwan have home defense centers affiliated with local governments, providing various resources needed by victims of domestic violence, sexual assault, and abuse, and also accepting 113 lines, police, campuses, and medical institutions’ notification. After receiving the notification, the home defense center will determine the priority of the incident and immediately provide emergency assistance such as assistance in reporting the case, accompanying medical treatment, or follow-up accompanying in court appearances, interrogation, and psychological counseling needs [22].

The results of this study found that the medical treatment rate of older adults who suffered from violence over the years increased significantly after standardization. Furthermore, the standardized medical treatment rate of the older adult generation after 2006 was even higher than that of the children and young generations. It consistently increased every year after 1959, and the increase was even greater after 2010 [23]. Advanced countries such as Canada also face the dilemma of an aging population and an increasing trend of older adult abuse [24]. In addition, the results of this study show that among the older adult victims of violence, the lower rate may be related to the proportion of the older adult population. In 2015 and 2020, the proportion of older adults over 65 years old in Taiwan was 12.5% and 16%, respectively [25]. Previous studies have highlighted that the loss of physical and mental function in older adults and needing care is one of the main risks of violence [26]. Therefore, with the accelerated aging of Taiwan’s population composition, the population exposed to the risk of violence in older adults is expected to increase and become more serious. Thus, the government should continue to pay attention to these issues.

Violence is caused by the presence of multiple risk factors and a combination of very few protective factors. Violence can be prevented by reducing risk factors and strengthening protective factors. Doing this requires comprehensive policies that form part of a so-called “integrated approach” to violence prevention—in other words, an overall strategy that depends on the cooperation of many different sectors. Violence prevention, therefore, requires interventions at all levels of the ecological model and every stage of the violence life cycle [22,27].

Critical to this is the timing of preventive interventions. Three key stages of prevention have been developed that have applications in various fields, from public health to violence prevention [27].

Stage 1: Primary Violence Prevention

Primary prevention aims to prevent violent behavior/activities from occurring. This type of intervention aims to address risk factors known to be associated with violence. Many of the risk factors and interventions discussed in Section 2 involve primary prevention activities [28].

Stage 2: Secondary Violence Prevention

Secondary prevention refers to any effort to intervene in populations already at high risk to ensure that violence does not occur. These include focusing on limiting environments conducive to violence (e.g., improving living standards through urban planning initiatives, providing recreational activities for violent youth, or providing emergency services), increasing the awareness of violence, people’s abilities (e.g., by providing counseling services to deal with intra-family conflict), and increased social cohesion [29].

Stage 3: Prevention of Tertiary Violence Tertiary prevention 

(a)providing long-term care after violence has occurred [30].(b)working to prevent the offender from relapsing. These can include any efforts to assist offenders with recovery and reduce recidivism, as well as efforts to provide support to victims, for example, by providing trauma counseling and other health-related services [30].

This study has several limitations. Firstly, this study can only initially analyze three generations of victims of medical treatment based on HWDC and temporarily cannot collect data on people who have not sought medical treatment for analysis. Therefore, the actual risk still needs further in-depth research. Second, potential influencing factors, such as other social supports and family status, should also be considered. Third, selection bias cannot be ruled out. This is because, in some cases, the victims may be reluctant to report violent incidents, and patients with minor injuries may not be admitted to the hospital. Fourth, the code given by clinicians will indeed affect the results of this study. Inevitably, the results observed in this study may still be underestimated. Finally, according to the relevant regulations of the HWDC, MOHW of Taiwan, we cannot obtain new analysis results in the short term, but even if we can successfully obtain new results, we could expect that the direction of the results will not change. After all, the main purpose of this study is to observe the trend of medical treatment of victims of violence in Taiwan over the years; strictly speaking, the data we use is a sample of 2 million people out of 23 million in Taiwan, which is what we can analyze. All the data, and the amount of such data is enough to represent the situation faced by the entire population of Taiwan.

## 5. Conclusions

The rate of medical treatment of adults suffering from violence has been decreasing in Taiwan from 2000 to 2015. With the accelerated aging of Taiwan’s population, it is expected that the Elderly exposed to the risk of violence will also increase and become more serious. The current problem of the declining birthrate and aging population will inevitably affect Taiwan’s entry into a super-aged society with an aging population in the near future. This may lead to the problem that the youngest adults in society are hostile to older adults. Social myths or stereotypes about older adults will also strengthen the self-social isolation of older adults. Therefore, how to help the “older adult person” accept and adapt to the “individual” aging? How to help the “family” accept and adapt to the aging of the “family?” and how to help “others” accept and adapt to the concept of “society” aging, and other concepts of “aging education” are pressing factors.

Future studies should investigate whether there have been any changes in the three generations, including violent crime, injuries, and rates of violence-related medical care, over the observation period from 2013 to 2022 (e.g., compared every ten years).

## Figures and Tables

**Figure 1 ijerph-19-07874-f001:**
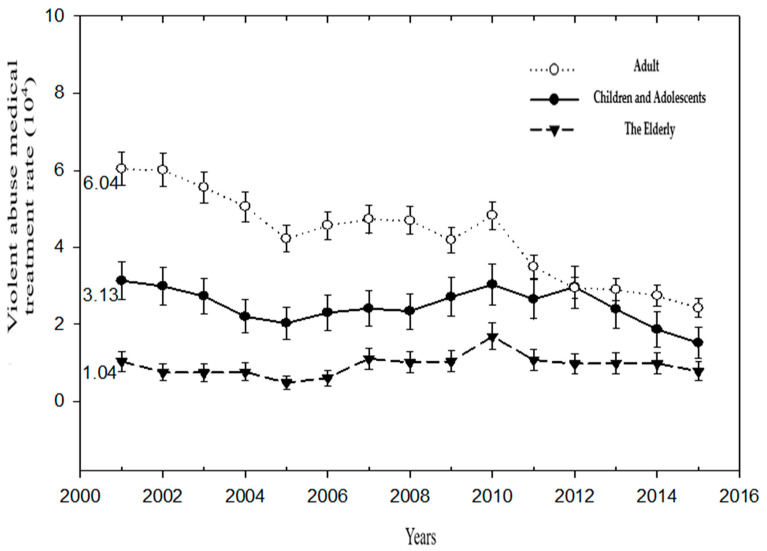
Crude rates of medical treatment over the years for victims of violence and abuse in three generations: Children and Adolescents, Adults, and The Elderly.

**Figure 2 ijerph-19-07874-f002:**
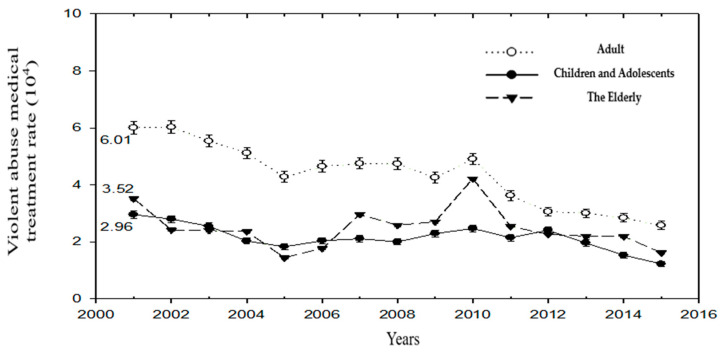
Standardized rates of initial violence-related medical visits among Children and Adolescents, Adults, and The Elderly.

**Figure 3 ijerph-19-07874-f003:**
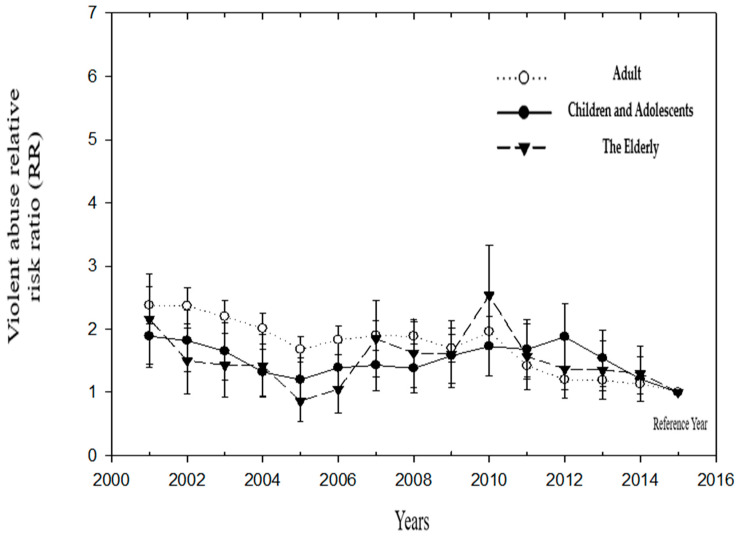
The relative risk ratio (RR) of medical treatment for violence in Children and Adolescents, Adults, and The Elderly over the years (Reference group: 2015).

**Figure 4 ijerph-19-07874-f004:**
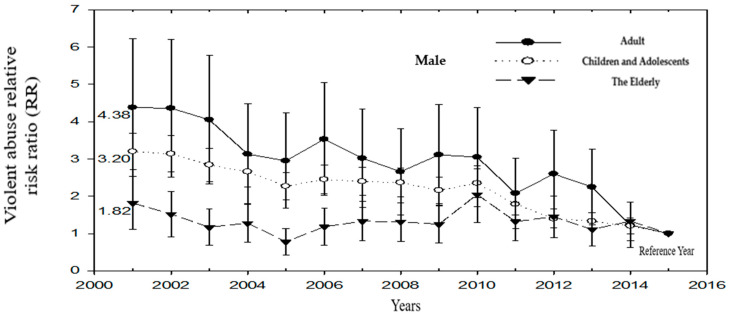
The relative hazard ratio (RR) of seeking medical treatment for violence in children, adults, and the elderly over the years (Male) (Reference group: 2015).

**Figure 5 ijerph-19-07874-f005:**
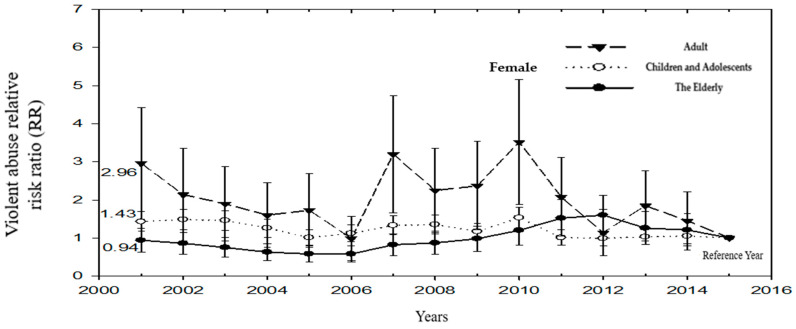
The relative hazard ratio (RR) of seeking medical treatment for violence in children, adults, and the elderly over the years (Female) (Reference group: 2015).

**Table 1 ijerph-19-07874-t001:** Crude rates of first-time medical treatment for Children and Adolescents, Adults, and The Elderly.

Year	Children and Adolescents (*n* =1 592)		Adults (*n* = 8726)		The Elderly (*n* = 759)	
	Crude Incidence	*95% CI*	Crude Incidence	*95% CI*	Crude Incidence	*95% CI*
2001	3.13	2.64–3.63	6.04	5.62–6.47	1.04	0.78–1.31
2002	2.99	2.51–3.48	6.00	5.57–6.42	0.75	0.53–0.97
2003	2.73	2.26–3.20	5.56	5.15–5.96	0.74	0.52–0.96
2004	2.20	1.77–2.62	5.06	4.67–5.44	0.76	0.53–0.99
2005	2.03	1.61–2.44	4.22	3.87–4.57	0.48	0.30–0.67
2006	2.30	1.85–2.74	4.57	4.21–4.93	0.61	0.41–0.82
2007	2.41	1.95–2.88	4.73	4.36–5.09	1.11	0.83–1.39
2008	2.34	1.88–2.80	4.69	4.33–5.05	1.01	0.74–1.28
2009	2.71	2.21–3.21	4.19	3.85–4.53	1.04	0.77–1.32
2010	3.03	2.49–3.57	4.83	4.47–5.20	1.69	1.34–2.04
2011	2.65	2.14–3.16	3.49	3.19–3.80	1.08	0.80–1.36
2012	2.97	2.42–3.51	2.94	2.66–3.22	0.97	0.70–1.24
2013	2.39	1.89–2.89	2.90	2.62–3.18	0.99	0.72–1.26
2014	1.87	1.42–2.31	2.74	2.47–3.01	0.98	0.71–1.25
2015	1.52	1.11–1.92	2.42	2.17–2.68	0.78	0.54–1.02

Denominator: The demographic structure of the Ministry of the Interior was converted to a population structure of 2 million people, and the population structure was not adjusted. Crude incidence: The unit is 1/10^4^.

**Table 2 ijerph-19-07874-t002:** Standardized rates of initial violence-related medical visits among Children and Adolescents, Adults, and The Elderly.

Year	Children and Adolescents (*n* = 1592)		Adults (*n* = 8726)		The Elderly (*n* = 759)	
	Normalized Rate	*95% CI*	Normalized Rate	*95% CI*	Normalized Rate	*95% CI*
2001	2.96	2.83–3.09	6.01	5.78–6.23	3.52	3.46–3.59
2002	2.80	2.68–2.93	6.03	5.81–6.26	2.42	2.37–2.47
2003	2.55	2.43–2.67	5.54	5.33–5.76	2.40	2.35–2.45
2004	2.03	1.92–2.13	5.12	4.92–5.33	2.36	2.31–2.41
2005	1.83	1.73–1.93	4.28	4.09–4.47	1.44	1.40–1.48
2006	2.04	1.94–2.15	4.65	4.45–4.84	1.77	1.73–1.82
2007	2.11	2.00–2.22	4.75	4.56–4.95	2.97	2.91–3.02
2008	2.00	1.90–2.11	4.74	4.54–4.93	2.59	2.54–2.64
2009	2.29	2.17–2.40	4.26	4.08–4.45	2.71	2.66–2.76
2010	2.47	2.35–2.59	4.91	4.71–5.11	4.22	4.16–4.29
2011	2.15	2.03–2.26	3.63	3.46–3.80	2.55	2.51–2.60
2012	2.41	2.29–2.53	3.06	2.91–3.22	2.26	2.22–2.31
2013	1.96	1.85–2.07	3.01	2.86–3.17	2.20	2.15–2.24
2014	1.53	1.43–1.63	2.85	2.70–3.00	2.19	2.15–2.23
2015	1.23	1.14–1.32	2.58	2.44–2.73	1.62	1.58–1.65

Denominator: The demographic structure of the Ministry of the Interior is converted into a population structure of 2 million people, and the age structure is corrected by the world standard population in 2000. Normalized rate: The unit is 1/10^4^.

**Table 3 ijerph-19-07874-t003:** Relative risk ratios (RR) of Children and Adolescents, Adults, and The Elderly who were subjected to violence to seek medical treatment over the years.

Year	Children and Adolescents (*n* = 1592)		Adults (*n* = 8726)		The Elderly (*n* = 759)	
	*RR*	*95% CI*	*RR*	*95% CI*	*RR*	*95% CI*
2001	1.89	1.39–2.58	2.38	2.09–2.70	2.16	1.45–3.21
2002	1.82	1.33–2.49	2.37	2.08–2.69	1.50	0.98–2.30
2003	1.65	1.20–2.27	2.20	1.94–2.50	1.43	0.93–2.20
2004	1.32	0.95–1.84	2.01	1.76–2.29	1.42	0.92–2.18
2005	1.20	0.86–1.68	1.68	1.47–1.92	0.86	0.53–1.40
2006	1.39	1.00–1.93	1.83	1.60–2.09	1.05	0.67–1.66
2007	1.43	1.03–1.99	1.90	1.67–2.16	1.85	1.24–2.74
2008	1.38	0.99–1.92	1.89	1.66–2.15	1.62	1.08–2.43
2009	1.58	1.14–2.18	1.70	1.48–1.94	1.61	1.08–2.42
2010	1.73	1.26–2.39	1.96	1.72–2.23	2.54	1.75–3.67
2011	1.68	1.21–2.34	1.42	1.24–1.63	1.57	1.05–2.35
2012	1.88	1.36–2.61	1.20	1.04–1.39	1.37	0.90–2.07
2013	1.54	1.10–2.15	1.19	1.03–1.37	1.35	0.89–2.03
2014	1.21	0.85–1.74	1.13	0.98–1.30	1.30	0.86–1.96
2015	1.00		1.00		1.00	

Analyzed by Poisson regression; 2015 was the reference group.

**Table 4 ijerph-19-07874-t004:** Relative hazard ratio (RR) of Children and Adolescents, Adults, and The Elderly who were subjected to violence to seek medical treatment over the years (Male).

Year	Children and Adolescents (*n* = 833)		Adults (*n* = 5853)		The Elderly (*n* = 477)	
	*RR*	*95% CI*	*RR*	*95% CI*	*RR*	*95% CI*
2001	4.38	2.54–7.57	3.20	2.71–3.78	1.82	1.11–2.97
2002	4.36	2.52–7.53	3.14	2.66–3.71	1.52	0.92–2.51
2003	4.05	2.33–7.02	2.84	2.40–3.36	1.17	0.69–2.00
2004	3.13	1.78–5.51	2.66	2.24–3.15	1.28	0.76–2.16
2005	2.95	1.67–5.20	2.27	1.90–2.70	0.78	0.43–1.43
2006	3.53	2.02–6.18	2.45	2.06–2.91	1.18	0.69–2.01
2007	3.02	1.71–5.33	2.40	2.02–2.85	1.33	0.80–2.22
2008	2.66	1.50–4.74	2.37	1.99–2.81	1.31	0.79–2.18
2009	3.11	1.76–5.48	2.16	1.81–2.57	1.25	0.75–2.09
2010	3.05	1.73–5.39	2.35	1.97–2.79	2.05	1.29–3.25
2011	2.08	1.13–3.80	1.79	1.50–2.15	1.32	0.80–2.18
2012	2.60	1.44–4.67	1.40	1.15–1.69	1.45	0.89–2.37
2013	2.25	1.23–4.11	1.33	1.10–1.61	1.11	0.66–1.86
2014	1.23	0.62–2.41	1.20	0.99–1.46	1.33	0.81–2.18
2015	1.00		1.00		1.00	

by Poisson regression; 2015 was the reference group.

**Table 5 ijerph-19-07874-t005:** Relative hazard ratio (RR) of Children and Adolescents, Adults, and The Elderly who were subjected to violence to seek medical treatment over the years (Female).

Year	Children and Adolescents (*n* = 759)		Adults (*n* = 2873)		The Elderly (*n* = 282)	
	*RR*	*95% CI*	*RR*	*95% CI*	*RR*	*95% CI*
2001	0.94	0.63–1.41	1.43	1.17–1.75	2.96	1.50–5.85
2002	0.86	0.57–1.30	1.48	1.21–1.81	2.14	0.93–4.91
2003	0.75	0.49–1.15	1.46	1.20–1.79	1.90	0.92–3.96
2004	0.63	0.41–0.99	1.26	1.02–1.55	1.60	0.75–3.40
2005	0.58	0.36–0.92	1.01	0.81–1.26	1.73	0.76–3.95
2006	0.58	0.37–0.93	1.12	0.90–1.39	0.99	0.41–2.39
2007	0.82	0.53–1.25	1.33	1.08–1.63	3.20	1.66–6.17
2008	0.87	0.57–1.33	1.35	1.10–1.65	2.25	1.14–4.45
2009	0.98	0.65–1.48	1.17	0.95–1.44	2.37	1.21–4.63
2010	1.20	0.81–1.79	1.53	1.26–1.87	3.52	1.88–6.62
2011	1.52	1.03–2.26	1.01	0.81–1.26	2.08	1.05–4.11
2012	1.60	1.08–2.37	0.99	0.80–1.23	1.14	0.53–2.45
2013	1.26	0.83–1.90	1.03	0.83–1.28	1.85	0.93–3.67
2014	1.21	0.79–1.84	1.05	0.85–1.30	1.45	0.69–3.05
2015	1.00		1.00		1.00	

by Poisson regression; 2015 was the reference group.

## Data Availability

Data are available from the NHIRD published by the Taiwan NHI administration. Due to the legal restrictions imposed by the government of Taiwan concerning the “Personal Information Protection Act,” data cannot be made publicly available. Requests for data can be sent as a formal proposal to the NHIRD (http://www.mohw.gov.tw/cht/DOS/DM1.aspx?f_list_no=812 (accessed on 13 January 2022)).

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
