# Peer review of "Trend Distribution of Violent Injuries in Taiwan from 2000 to 2015"

_ijerph, 2022, doi:10.3390/ijerph19137874_

Round 1

Reviewer 1 Report

This article is interesting and well constructed. Some details are necessary in the method and results section to better understand this study. My comments are in the PDF

Author Response

See attach file.

Reviewer 2 Report

Using the National Health Insurance Research Dataset, this study analyses analysed violent injuries in Taiwan for the period between 2000 and 2015. The topic is interesting although the following issues need to be clarified.

1.      The use of poisson regression model needs to be clarified: why other advanced models, such as random-parameters count models that may identify unobserved heterogeneity, were not considered.

2.      E-codes in emergency rooms are in general not reliable, primarily because ER physicians seldom E-codes. Please clarify how this is likely to impact the results/conclusions drawn.

3.      Severity of violence injuries should have been analysed.  Analysing such data, that can be obtained from ICD-9/10, may contribute to the literature more significantly than those identified in the current study.

4.      The contribution of this study to relevant literature is limited by the fact that most of findings have already documented in literature. Please consider focusing on specific groups such as injuries from child abuse, intimacy injuries, etc.

Author Response

See attach file.

Reviewer 3 Report

Thank you for your work on "Trend distribution of violent injuries in Taiwan from 2000 to 2015".

Introduction:

Regarding the last paragraph - there is published work on violence (workplace violence) in Taiwan.

See:

1. Han, C. Y., Lin, C. C., Barnard, A., Hsiao, Y. C., Goopy, S., & Chen, L. C. (2017). Workplace violence against emergency nurses in Taiwan: A phenomenographic study. Nursing outlook65(4), 428-435.

2. Lee, H. L., Han, C. Y., Redley, B., Lin, C. C., Lee, M. Y., & Chang, W. (2020). Workplace violence against emergency nurses in Taiwan: a cross-sectional study. Journal of emergency nursing46(1), 66-71.

There might be other related published work. I suggest authors should cite the available related literature in the introduction especially studies conducted in similar geographical areas.

Regarding the last sentence in the introduction, what does the term 'rough rate of medical abuse' mean? It wasn't explained in the manuscript.

Methods:

Section 2.2: Is there any reason why "victims of violence" is written twice?

Also, how the rates - 'rough' and 'standardized' are calculated is not explained.

Discussion:

The first sentence should specify the years of reference for the study.

Kindly cite the source(s) for the stages of prevention and related studies (if any) that might have used these stages of prevention as interventions in preventing violent injuries.

Conclusion:

Kindly review/ reword your first sentence to remove any insinuation of causality because this is a retrospective study using secondary data.

Thank you.

Results:

Gross medical treatment rate is not defined.

Author Response

See attach file.

Round 2

Reviewer 1 Report

thanks to the authors for the additions and corrections. I have no more comments for this article.

Author Response

See attachment flie.

Reviewer 2 Report

Most of my previous comments and questions are not addressed sufficiently. In particular, advanced count models such as random-parameters count models should be estiimated.   This reviewer cannot be more positive on this paper and has to reject this paper. 
